# Gross Chromosomal Rearrangements in *Kluyveromyces marxianus* Revealed by Illumina and Oxford Nanopore Sequencing

**DOI:** 10.3390/ijms21197112

**Published:** 2020-09-26

**Authors:** Lin Ding, Harrison D. Macdonald, Hamilton O. Smith, Clyde A. Hutchison, Chuck Merryman, Todd P. Michael, Bradley W. Abramson, Krishna Kannan, Joe Liang, John Gill, Daniel G. Gibson, John I. Glass

**Affiliations:** 1J. Craig Venter Institute, 4120 Capricorn Lane, La Jolla, CA 92037, USA; linding@usc.edu (L.D.); harrison.macdonald01@gmail.com (H.D.M.); hsmith@jcvi.org (H.O.S.); chutchison@jcvi.org (C.A.H.III); chuckmerryman@gmail.com (C.M.); toddpmichael@gmail.com (T.P.M.); theabramson@gmail.com (B.W.A.); dan@codexdna.com (D.G.G.); 2Codex DNA, 9535 Waples St #100, San Diego, CA 92121, USA; krishna@codexdna.com (K.K.); madonjoe@gmail.com (J.L.); john@codexdna.com (J.G.)

**Keywords:** gross chromosomal rearrangements, non-homologous end joining, translocation, Illumina MiSeq, Oxford Nanopore, *Kluyveromyces marxianus*, *Saccharomyces cerevisiae*, *URA3* gene

## Abstract

*Kluyveromyces marxianus* (*K. marxianus*) is an increasingly popular industrially relevant yeast. It is known to possess a highly efficient non-homologous end joining (NHEJ) pathway that promotes random integration of non-homologous DNA fragments into its genome. The nature of the integration events was traditionally analyzed by Southern blot hybridization. However, the precise DNA sequence at the insertion sites were not fully explored. We transformed a PCR product of the *Saccharomyces cerevisiae URA3* gene (*ScURA3*) into an uracil auxotroph *K. marxianus* otherwise wildtype strain and picked 24 stable Ura+ transformants for sequencing analysis. We took advantage of rapid advances in DNA sequencing technologies and developed a method using a combination of Illumina MiSeq and Oxford Nanopore sequencing. This approach enables us to uncover the gross chromosomal rearrangements (GCRs) that are associated with the *ScURA3* random integration. Moreover, it will shine a light on understanding DNA repair mechanisms in eukaryotes, which could potentially provide insights for cancer research.

## 1. Introduction

*Kluyveromyces marxianus* is a thermotolerant yeast [1] and is the fastest-growing eukaryote identified to date [2]. It has many other physiological features that the conventional yeasts, such as *Saccharomyces cerevisiae*, are lacking. As a result, *K. marxianus* is becoming a potentially valuable industrial yeast. Therefore, many molecular tools [3,4,5,6,7] have been developed for its genetic engineering. Among these are many that harness the power of the homologous recombination (HR) pathway, which requires homologous flanking sequences, for targeted gene editing. However, similar to its cousin, *Kluyveromyces lactis* [8], *K. marxianus* also embraces a robust non-homologous end joining (NHEJ) pathway in the absence of homology. It can be transformed with non-homologous DNA fragments by illegitimate recombination (IR) that involves the NHEJ pathway [9]. In the resulting transformants, the DNA fragments insert at various chromosomal locations. These two modes exist with various frequencies in different yeast strains [10,11].

Kegel et al. [8] reported genome-wide IR insertion events in *Kluyveromyces lactis* that were strongly biased toward intergenic regions. If they induced ectopic double strand breaks (DSBs) using restriction cleavage, then there was no bias for genic versus intergenic sequence, suggesting that genome-wide IR occurred at spontaneous mitotic DSBs that are preferentially distributed in intergenic sites. Nonklang et al. [12] used a PCR product of the *Saccharomyces cerevisiae URA3* (*ScURA3*) gene to transform a *K. marxianus* DMKU3-1042 *ura3∆* mutant. They analyzed 9 Ura+ transformants by Southern blot hybridization. The *ScURA3* insertions were in different genome locations and one transformant had multiple insertions. Abdel-Banat et al. [9] reported very high frequency insertion of *ScURA3* DNA into the genome of *K. marxianus* DMKU3-1042 (2×10^6^ transformants/µg DNA) by the NHEJ pathway. They suggested that a high-density *ScURA3* insertion map analogous to that obtained by global transposon mutagenesis [13,14] could be generated. In order to fully understand the NHEJ repair mechanism in *K. marxianus*, it is desired to have the precise sequence at the junctions of insert sites in the genome. This is not possible to be achieved by Southern blot hybridization due to its low resolution, not to mention its laboriousness.

We generated our own Ura+ *K. marxianus* transformants by transforming a 1123-bp PCR product of the *Saccharomyces cerevisiae URA3* gene (*ScURA3*) and took advantage of Illumina MiSeq and whole-genome Oxford Nanopore sequencing to produce a detailed analysis of 24 Ura+ transformant clones. We found a surprising variety of insertion events. In addition to three tandem dimer and two trimer *ScURA3* concatemer insertions, two insertions produced inversions, one produced a large deletion, and another three Ura+ transformants produced chromosomal translocations in which each end of *ScURA3* were inserted in a different chromosome. Sequencing analysis, especially the long reads obtained with Nanopore sequencing, was instrumental in detecting and analyzing GCRs that are a hallmark of cancers. Therefore, our results and method may provide insights to understanding the basic mechanisms of DNA repair and cancer biology.

## 2. Results

### 2.1. Determination of ScURA3 Genomic Insertion Sites by Illumina MiSeq Analysis

We constructed two separate libraries containing either *ScURA3* head or tail junctions (Figure 1). Illumina MiSeq 100-bp sequences from the head library were scanned for *ScURA3* head sequence matches followed by at least 20 bp of the *K. marxianus* sequence. This *K. marxianus* sequence identified the chromosome and the site of a *ScURA3* insertion. Similarly, *ScURA3* tail junctions were identified. In total, we identified 23 head and 23 tail genome junctions (Figure 2). If head and tail junctions were very close on the same chromosome, they were paired and assumed to be produced by a single *ScURA3* insertion event. This could then be confirmed by PCR using primers designed from the flanking *K. marxianus* sequences (Appendix A). Testing all the clones with a given primer set will point to the specific clone carrying the *ScURA3* insert.

### 2.2. Concatemer Dimer and Trimer ScURA3 Inserts

The 24 clones were screened for the presence of URA3+ concatemers by performing PCR with the primers ura3+61c and ura3+720 (Appendix A). If *ScURA3* head to tail joints are present, a PCR product of 463 bp should result. Five clones 2, 3, 5, 12, and 19 gave the expected product. In addition, PCRs performed with primers flanking the *ScURA3* concatemers yielded products of the expect sizes. Clones 3, 5, and 19 contained dimers and clones 2 and 12 contained trimers. These PCRs also showed ladders of bands corresponding to monomers, dimers, and higher bands as expected since during the annealing step of each PCR cycle, some of the *ScURA3* concatemers may hybridize out of phase (Appendix A). These results were subsequently confirmed by Oxford Nanopore sequence reads that spanned the entire concatemer and flanking sequence.

The head to tail junctions in the dimer and trimer inserts might be expected to show the effects of NHEJ repair. These junctions are readily observed in both Oxford Nanopore and Illumina MiSeq reads. Among the seven junctions, we observed only three different types: CCCTG|TGAGA, CCC***tg***|TGAGA, and CC***A***T***g***|TGAGA, where the bold lowercase italicized letters are deleted bases and the bold uppercase italicized letters are substitutions or new bases. Note that we cannot distinguish whether the ***tg*** in the second junction type is deleted from the tail as shown, or from the head.

### 2.3. Sixteen ScURA3 Insertion Events Were Either Precise or Resulted in Small Deletions of Genome Sequence

Sixteen clones appeared to be simple events that involved insertion of *ScURA3* into a single break in the *K. marxianus* genome (Table 1). Two of the clones (3 and 17) contained precise *ScURA3* insertions, that is, no loss of genome sequence occurred at the insertion site. In both cases, the *ScURA3* terminal head sequences were unaltered; however, there was a loss of the terminal G of the tail sequence in both cases. In 14 clones (2, 4, 9, 10, 11, 12, 14, 16, 18, 19, 20, 21, 22, and 24), insertions were accompanied by small deletions of the genome sequence, ranging from 2 to 30 bp. Several of these also involved alterations of the terminal head or tail sequences at the junctions. In clone 19, the *ScURA3* tail had lost 24 terminal bases followed by 4 base substitutions (Figure 2). The observations are compatible with DSB repair by the error-prone NHEJ pathway [15].

### 2.4. Two ScURA3 Insertion Events Were Simple Insertions but Involved Corruption of the ScURA3 Termini

Clone 1 has an insertion in the ade1p gene at position 397,946 on chromosome 1, which is in the open reading frame of (*SG4EUKG585063*, homologous to *ScADE1*). As a result of the disruption of the *ADE1* gene, *K. marxianus* colonies are pink on low adenine medium. This is consistent with Ade mutants in other yeasts. It appeared to be precise except for an inserted 65 bp DNA sequence between the *ScURA3* tail junction and the genome sequence. This DNA did not convincingly match any of the genome sequence. Its source is not known. It seems unlikely, because of its length, to have been inserted by an NHEJ polymerase not requiring a template. One result of this insertion was the inability to identify a tail junction sequence for clone 1 in the MiSeq data. The head junction sequence was readily identified. Only by Oxford Nanopore sequencing was the insertion detected and the point of insertion of *ScURA3* DNA into the genome identified.

In clone 13, about 20 bp of the *ScURA3* tail sequence and 78 bp of the head sequence were missing, thus there were no junction sequences in Figure 2 and the chromosome location of the insert was not found.

### 2.5. Three ScURA3 Insertions Resulted in Large Chromosomal Inversions or Deletions

Clone 6 produced an inversion of the genomic segment between 2 DSBs 47,986 bp apart on chromosome 4 (Table 1 and Figure 3). Oxford Nanopore reads were necessary to solve the structure (Appendix A). Each DSB produces two ends, and these can be labeled 1 and 2 for the first break, and 3 and 4 for the second break (Figure 3A). *ScURA3* DNA has two ends labeled H and T. Thus, several possible repair reactions can occur. Clone 6 exhibited 1H–T3 joining followed by 2–4 joining, resulting in inversion of the 2–3 segment. Contour-clamped homogeneous electric field (CHEF) electrophoresis gel analysis of this clone was identical to the wild type (Figure 4).

Clone 7 involved 2 DSBs in chromosome 3 and one DSB in chromosome 7. First, a 129,349 bp 2—3 segment was deleted from chromosome 3 in a 1T—H4 *ScURA3* insertion reaction. This was followed by insertion of the 2—3 segment at a third DSB in chromosome 7 (Table 1, Figure 3A, and Appendix A). Thus, it is predicted that chromosome 7 will increase from 940 to 1070 kb and chromosome 3 will decrease from 1588 to 1458 kb as confirmed by CHEF gel analysis (Figure 4).

The clone 8 insertion is more complex. In this case, the DSBs were 1893 bp apart on chromosome 1 and *ScURA3* integration involved 2T—H4 joining, resulting in an inversion of the 1893-bp segment. The 1 and 3 ends then interacted with another *ScURA3* sequence, yielding the inverted 1892-bp segment separating the two *ScURA3s*. Furthermore about 200 bp of the head of the second *ScURA3* is missing. The overall structure is thus 1T—ΔH3—2T—H4 (Figure 3A). Several Oxford Nanopore reads confirm this structure both from the 24-clone pool and from the barcoded clone 8 reads (Appendix A). The CHEF gel chromosome banding pattern was wild type as expected (Figure 4).

### 2.6. Three ScURA3 Insertions Resulted in Inter-Chromosomal Translocations

For clones 5, 15, and 23, Oxford Nanopore reads located the *ScURA3* head and tail junctions on separate chromosomes, indicating translocation events (Table 1, Appendix A). In clone 5, a *ScURA3* dimer inserted to produce a reciprocal exchange between the arms of chromosomes 7 and 6 (Appendix A) as diagrammed in Figure 3B. One pair of arms is bridged by the *ScURA3* dimer. The nature of the rejoin by the other two arms has not been determined. The CHEF gel bands are similar to wild type (Figure 4).

Clone 15 contained a *ScURA3* monomer insert with one end in chromosome 1 and the other in chromosome 5 (Appendix A). The *ScURA3* tail junction in chromosome 5 falls in a region of repetitious sequence making the exact location uncertain. However, *K. marxianus* assembly based on Oxford Nanopore reads gives a single location in chromosome 5. The rejoin of chromosome 1 and 5 was not successfully located in the Oxford Nanopore reads. Chromosome 5 is predicted to decrease from 1353 to 1279 kb while chromosome 1 should increase from 1745 to 1819 kb (Figure 4).

Clone 23 contained a *ScURA3* insert with one end in chromosome 4 and the other in chromosome 6. The translocation is interesting in that the short ends of chromosome 4 and 6 are joined to yield a new short chromosome (Figure 3B and Appendix A). Rejoin of the long arms occurs at positions 7293 and 103467 to yield an extra-long chromosome. In addition, a short piece of chromosome 4 is released and integrates into chromosome 5 at approximately position 627037 (Figure 3B). Bands 4 and 6 disappear, while the short and very long new chromosomes are not identifiable on the CHEF gel (Figure 4). Interestingly, the band for chromosome 5 also seems to have disappeared.

## 3. Discussion

During transformation of *K. marxianus*, there is typically a 50- to 100-fold excess of *ScURA3* DNA molecules to yeast cells. It is expected that many cells will take up several *ScURA3* molecules. The NHEJ pathway proteins, Ku70 [9], and Ku80 [6] are required for production of Ura+ in *K. marxianus*. NHEJ proteins act on the free *ScURA3* ends, resulting in monomer circles as well as tandem concatemers. The probability of these different events is not known, but it seems likely that free unreacted *ScURA3* ends would not persist for long. Proximity of ends to each other probably determines how likely a pair of ends are to react. However, if endogenous DSBs are simultaneously present in the genome of the same cell, these would be expected to occasionally join to free *ScURA3* ends to yield Ura+ transformants. This is a rare event leading to only a few thousand transformants among the millions of *K. marxianus* cells present per transformation reaction. By plating on CAA-U plates, insertion of *ScURA3* DNA into the genome is specifically selected for since non-inserted *ScURA3* DNA does not independently replicate and is diluted out among the progeny cells. To discover the types of insertion events that might occur, we isolated 24 Ura+ clones for detailed study.

We developed a method using a combination of Illumina MiSeq and Oxford Nanopore sequencing analysis to reveal the precise nucleotide sequence right at the junction of *ScURA3* random insertion sites, instead of using the laborious Southern blot hybridization. We found that for certain insertions, only long-read Nanopore sequencing is capable of resolving the new structure. Our analysis showed that three clones contained dimer inserts and two were trimers. The rest were monomer inserts. Eighteen events involved simple insertion into a single DSB in the *K. marxianus* genome while six involved two or more DSBs. Three events produced translocations in which the two ends of the *ScURA3* cassette inserted into different chromosomes. Two of the latter events produced inversion of DNA between the two DSBs and one produced a large deletion in which the 129-kb deleted segment reinserted into another chromosome. Interestingly, no notable growth defects were observed in these clones even for clone 7 that lost 129 kb of genetic material. This indicates the elasticity that *K. marxianus* genome has. Further investigation may uncover DNA repair mechanisms that could be valuable to understanding chromosome evolution and therefore provide insight to speciation and cancer biology

## 4. Materials and Methods

### 4.1. Yeast Strains

*K. marxianus* NRRL Y-6860 was obtained from the U.S. Department of Agriculture Agricultural Research Service Culture Collection. We sequenced the genomic DNA and identified 8 chromosomes comprising the 10,837,618-bp genome, and 4963 genes were identified (GenBank number GCA_002356615.1). Strain G13 (*ura3∆*) was constructed as in Appendix A. The *KmURA3* ORF in *K. marxianus* NRRL Y-6860 was removed by homologous recombination (HR)-mediated 5-FOA counter selection. Primers used for strain construction and confirmation are listed in Appendix A.

### 4.2. Preparation of ScURA3 Cassette DNA

The *ScURA3* cassette contained in the plasmid pRS316 (ATCC^®^ 77145™; Manassas, VA, USA) was PCR-amplified using the two primers 5′-tgagagtgcaccacgcttttcaattc and 5’-cagggtaataactgatataattaaattg. The 5´ OH PCR product (1123 bp, Appendix A) was purified using the QIAquick PCR purification kit (Qiagen, Hilden, Germany). For purposes of calculation, 1 µg of *ScURA3* DNA contains approximately 10^12^ molecules. For convenience, the 5′ end of *ScURA3* is called the “head” and the 3′ end is the “tail” (Figure 1).

### 4.3. Isolation of K. marxianus ScURA3 Transformants and Preparation of Transformant DNAs

Transformation buffer (TFB) consists of 9 parts PEG/Li acetate solution (20 mL of 60% polyethylene glycol 3350 (MilliporeSigma 202444, Burlington, MA, USA), 1.5 mL of 4M lithium acetate, and 5.5 mL of sterile water) and 1 part of fresh 1M dithiothreitol. For transformation, *K. marxianus* cells were grown for 24 h in 30 mL of YPD medium at 30 °C, centrifuged at 3000 rpm for 5 min and resuspended in 900 µL of TFB. The cell suspension was transferred to a 1.5-mL Eppendorf tube and centrifuged at 3000 rpm for 5 min. The supernatant was removed, and the cells were resuspended in 600 µL of TFB. *ScURA3* DNA (70 ng) was then mixed with 50 µL of the *K. marxianus* cell suspension (containing approximately 10^9^ cells) and incubated at 42 °C for 30 min. In total, 100 µL of CAA-U medium (2% glucose, 0.6% casamino acid, 25 μg/mL adenine 50 μg/mL, tryptophan, and 0.67% YNB without amino acids) were added and the cells were spread on a CAA-U/2% agar plate and incubated at 30 °C for 2 days. Then, 24 isolated colonies were picked and patched onto a YPD plate. After incubation for a day at 30 °C, the 24 patches were re-patched on a CAA-U plate and grown another day. Twenty-four 50-mL tubes containing 10 mL of CAA-U medium were inoculated from the patches and grown at 30 °C for 24h on a shaker. Cells were harvested from each of the cultures by centrifugation and resuspended in 200 µL of P1 cocktail (5 mL of P1 solution (Qiagen, Hilden, Germany), 5 µL of 14M β-mercaptoethanol, and 125 µL of Zymolyase 20mg/mL) in 1.5-mL Eppendorf tubes. The cells were incubated at 37 °C for 30 min followed by addition of 20 µL of 3M sodium acetate, and extraction with an equal volume of phenol. After centrifugation, the supernatants were harvested and precipitated with 2 volumes of ethanol. The precipitates were washed with ethanol and dissolved in 200 µL of TE buffer (10 Mm Tris-Cl, 1 mM Na EDTA, pH 8). The 24 transformant DNAs ranged in concentration from approximately 50 ng/µL to 200 ng/µL.

In addition to the individual transformant DNAs, cells were pooled from a plate of 24 patches and extracted as above to yield 24-clone pool DNA at approximately 200 ng/µL.

### 4.4. Preparation of 24-Clone Pool DNA Libraries for Illumina MiSeq Sequencing

The 10.8 Mb *K. marxianus* genome contains approximately 37,500 Sau3AI restriction sites (5′GATC) occurring on average every 280 bp along the genome. There is a single site in the 1123-bp *ScURA3* sequence (Appendix A) that cleaves the sequence into a 918-bp left fragment and a 203-bp right fragment. The 24-clone pool DNA was digested with Sau3AI to produce fragments with 5′GATC overhangs in a reaction mixture (50 µL) containing 5 µL of 10X NEB 1.1 buffer (New England Biolabs, Ipswich, MA), 5 µL of 24-clone pool DNA (~200 µg/µL, ~10^8^ genome equivalents), 2 µL Sau3AI (10 ug/µL), and 38 µL water. Incubation was at 37 °C for 2h followed by inactivation of Sau3AI enzyme at 65 °C for 30 min. A 5-µL aliquot of the fragments was mixed with 20 µL of 10× T4 ligase buffer (NEB), 5 µL T4 ligase (400 ug/µL, New England Biolabs, Ipswich, MA), and 170 µL water. Incubation was at 23 °C for 17 h followed by 72 °C for 10 min to inactivate the ligase. Two PCR reactions were performed. The first contained 5 µL of the ligated DNA, 16 µL of water, 25 µL of 2× Q5 master mix (New England Biolabs, Ipswich, MA), and 2 µL of each of the primers ura3 + 720c and ura3 R1-1-DN at 25 µM. The second PCR was the same except that the primers were ura3 + 61c and ura3 TestF (Appendix A). PCR settings were 98 °C 10 s, 55 °C 20 s, and 72 °C 2 min for 30 cycles. The first PCR reaction contained *ScURA3* head to *K. marxianus* genome junctions and the second contained the tail junctions (Figure 1). PCR product yields were 83 and 109 ng/µL, respectively. Illumina MiSeq 100 nucleotide sequencing reads were done on the two libraries.

### 4.5. Oxford Nanopore Sequencing

Oxford Nanopore DNA sequencing of the 24-clone pool DNA was performed as described by Oxford Nanopore Technologies. It yielded 2.1 Gb of sequence with a mean read length of 4658 bp and a maximum read length of 194,297 bp (Appendix A). Single Nanopore reads gave a 10–15% base calling error rate including base deletions. However, alignment of multiple Nanopore reads and *K. marxianus* genome assembly yielded the sequence that was > 95% accurate compared to the MiSeq assembly. We generally relied on Illumina MiSeq data for design of primers.

In addition to Oxford Nanopore sequencing of the 24-pool DNA, each clone DNA was bar-coded and sequenced in two 12-clone pools. Reads for each clone were then collected into 24 files ranging from 21 to 827 Mb in size and averaging 370 Mb per clone.

### 4.6. CHEF Genomic DNA Plug Preparation

A single colony from Clone 5, 6, 7, 8, 13, 15, and 23 out of the 24 clones was inoculated into 50 mL YPD and grown at 30 °C overnight. The 1% agarose plugs were prepared using CHEF Yeast Genomic DNA Plug Kit (Bio-Rad170-3593, Hercules, CA, USA) following the manufacturer’s manual. The plugs were inserted into wells of 1% 0.5X TBE agarose gel and sealed with 1% 0.5× TBE agarose.

### 4.7. Pulsed-Field Gel Electrophoresis (PFGE)

CHEF-DR^®^ III Pulsed Field Electrophoresis Systems (Bio-Rad, Hercules, CA, USA) was used for running PFGE with the following settings: initial switch time: 26.3 s, final switch time: 228 s, gradient: 6 V/cm, angle: 120°, temperature: 14 °C, and total time: 36 h. Gel was stained with 0.5 µg/mL ethidium bromide solution in water for 30 min and de-stained in distilled water for 1 h. Genomic DNA was visualized using Typhoon 9410 Variable Mode Imager.

## Figures and Tables

**Figure 1 ijms-21-07112-f001:**
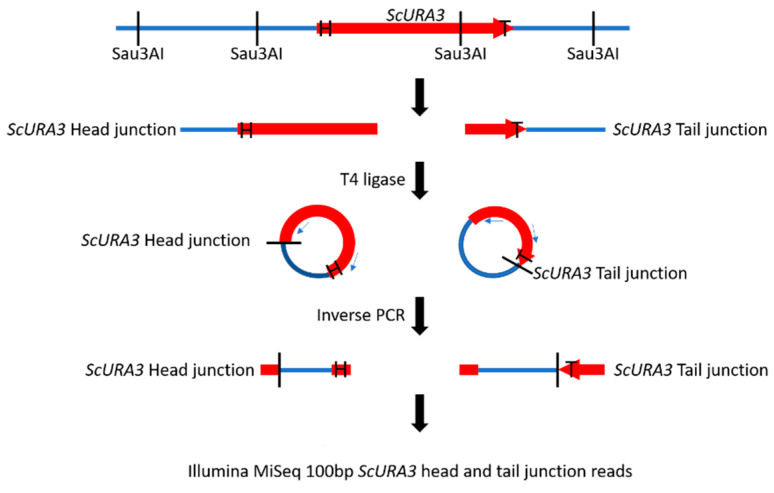
Construction of *ScURA3* head (arrow, labeled H) and tail (labeled T) junction libraries. DNA from a *ScURA3* transformant clone is cleaved at GATC sites with Sau3AI and the fragments are ligated under dilute conditions to yield DNA circles with separate *ScURA3* head and tail junctions. The ligated DNA is then divided into two aliquots. Circles containing the head junctions are amplified in a PCR reaction with primers ura3 + 61c and ura3 TestF. The tail junctions are amplified with ura3 + 720c and ura3 R1-1-DN primers. The result is separate head and tail junction libraries.

**Figure 2 ijms-21-07112-f002:**
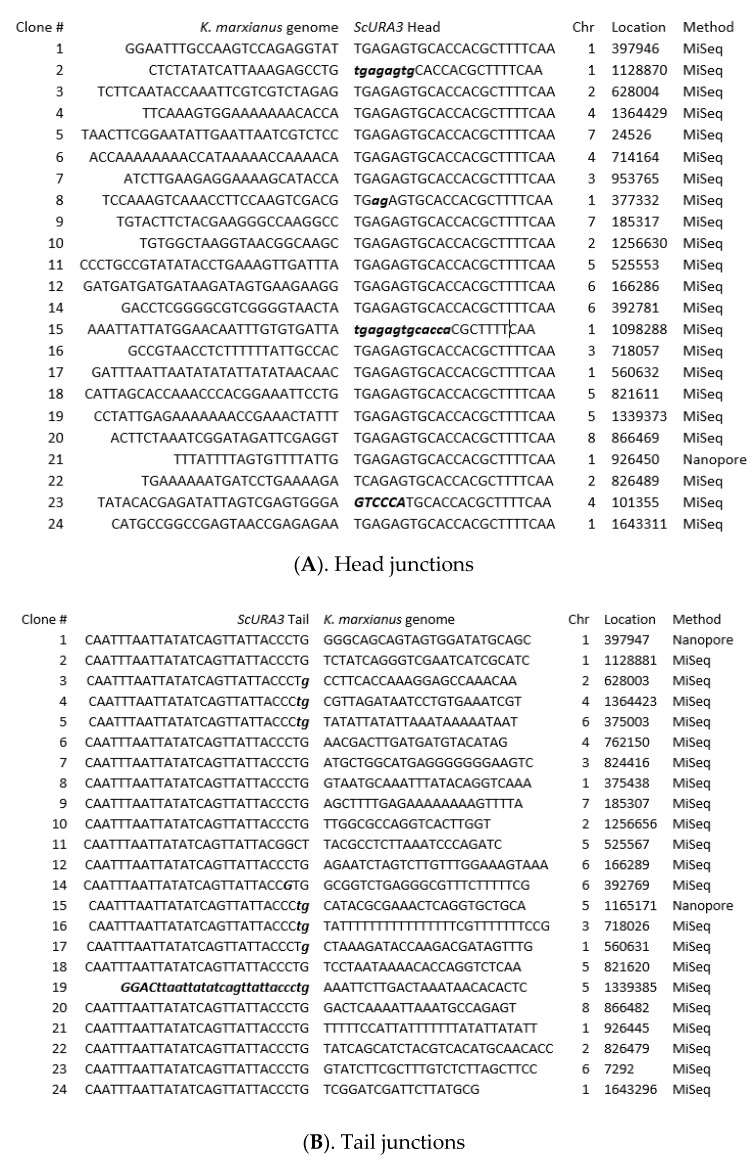
Head and tail junction sequences from Illumina MiSeq reads of the inverse PCR 24-clone libraries constructed as shown in Figure 1. (**A**). Head junctions. (**B**). Tail junctions. Chromosome (Chr) and insertion sites are indicated. Lower case, italicized, bold letters indicate deleted bases. Upper case, italicized, bold letters indicate base substitutions.

**Figure 3 ijms-21-07112-f003:**
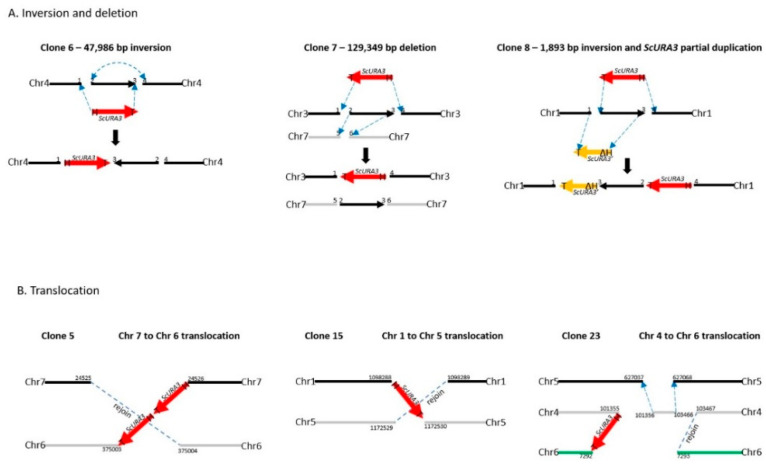
Diagrams of the structures of the *ScURA3* insertion events revealed from Illumina MiSeq and Oxford Nanopore sequencing analysis. (**A**). Inversion and deletion events observed in clone 6, 7, and 8. (**B**) Translocation events observed in clone 5, 15, and 23.

**Figure 4 ijms-21-07112-f004:**
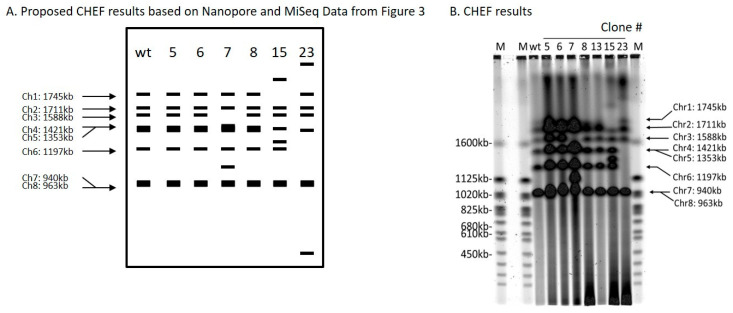
Contour-clamped homogeneous electric field (CHEF) analysis of *ScURA3* inversion and deletion as well as translocation events. (**A**). A schematic of expected band patterns of each clone. (**B**). CHEF results of clone 5, 6, 7, 8, 13, 15, and 23.

**Table 1 ijms-21-07112-t001:** Types of *ScURA3* insertion events. The asterisk next to clones 1 and 13 indicates incomplete information (see text). Chr: Chromosome. #: number (see text). ?: unknown.

Clone #	Primer Set	Chr	Head Junction Location	Tail Junction Location	Type of Insertion and Base Pair Involved		*ScURA3*	Gene or Intergenic Insertion
1 *	none	1	397946	397947	**precise**	**0 bp**	monomer	*SG4EUKG585063* (*ADE1*)
3	2	2	628004	628003	**precise**	**0 bp**	dimer	*SG4EUKG585526*
17	16	1	560632	560631	**precise**	**0 bp**	monomer	intergenic
2	1	1	1128870	1128881	**del**	**10 bp**	trimer	intergenic
4	26	4	1364429	1364423	**del**	**5 bp**	monomer	*SG4EUKG587507*
9	37	7	185317	185307	**del**	**9 bp**	monomer	intergenic
10	11	2	1256630	1256656	**del**	**25 bp**	monomer	intergenic
11	21	5	525553	525567	**del**	**13 bp**	monomer	intergenic
12	4	6	166286	166289	**del**	**2 bp**	trimer	intergenic
13 *	none	?	?	?	**?**	**?**	monomer	?
14	20	6	392781	392769	**del**	**11 bp**	monomer	intergenic
16	25	3	718057	718026	**del**	**30 bp**	monomer	intergenic
18	27	5	821611	821620	**del**	**8 bp**	monomer	*SG4EUKG588009*
19	3	5	1339373	1339385	**del**	**11 bp**	dimer	intergenic
20	chr8	8	866469	866482	**del**	**12 bp**	monomer	*SG4EUKG589280*
21	32	1	926450	926445	**del**	**4 bp**	monomer	intergenic
22	12	2	826489	826479	**del**	**9 bp**	monomer	*SG4EUKG585988*
24	17	1	1643311	1643296	**del**	**14 bp**	monomer	intergenic
6	9	4	714164	762150	**inversion**	**47,986 bp**	monomer	intergenic-intergenic
7	7	3	953765	824416	**del**	**129,349 bp**	monomer	intergenic-intergenic
8	5	1	377332	375438	**inversion**	**1892 bp**	monomer	*SG4EUKG584656*-intergenic
5	cl5	7–6	24526(7)	375003(6)	**Translocation**	**Translocation**	dimer	intergenic-intergenic
15	10	1–5	1098288(1)	1165171(5)	**Translocation**	**Translocation**	monomer	intergenic-intergenic
23	8	4–6	101355(4)	7292(6)	**Translocation**	**Translocation**	monomer	*SG4EUKG586913*-intergenic

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
