# Peer review of "Gross Chromosomal Rearrangements in Kluyveromyces marxianus Revealed by Illumina and Oxford Nanopore Sequencing"

_ijms, 2020, doi:10.3390/ijms21197112_

Round 1

Reviewer 1 Report

Ding et al. report the random integration sites of ScURA3 cassette in the genome of Kluyveromyces marxianus. The authors employ Illumina MiSeq and Oxford Nanopore technology to define the junctions of the integration sites in 24 Ura+ clones. Authors identified not only simple integration events but also defined chromosomal rearrangements triggered by the integration. Because these are random integration events, most of the junction sequences support non-homologous end joining (NHEJ) events.

The experimental design was somewhat insufficient to understand the full scope of the events. In particular, Nanopore results are not adequately presented. This reviewer would assume this is because of the lack of read depth of the Nanopore data. On average, 370Mb is read for each clone, which is approximately 26 fold coverage of the K marxianus genome (14Mb, which was not mentioned in the manuscript). With the high error rate, 26x coverage is not sufficient to assemble the individual clones’ genome, and thus to reveal a full scope of insertion events.

Each clone is supposed to have a different number of integration events, which should be able to find out by estimating the copy numbers of Ura3 cassette. This can be done either by Southern blot or quantitative digital PCR. Then the authors could use Illumina short read paired-end sequencing for the entire genome (not for PCR products) and Nanopore sequencing for some of the clones that have high incidences of integration. In this way, authors could have had sufficient read depths for each sequencing platform and revealed a full scope of the events, which is necessary for publication.

In addition to the insufficient presentation of the Nanopore sequencing data, the CHEF data is confusing. Is this Southern blot with the ScUra3 as a probe, or a simple CHEF gel picture? If it is a simple CHEF gel picture, why some clones lack some of the chromosomes? Instead of proposing an interpretation, authors could blot the gel into the membrane and probe with ScUra3. In this way, the authors will know the aberrant chromosomes are indeed induced by the integration events. Additional whole genome sequencing and copy number analysis could tell whether there are deletions and amplifications that contribute to the aberrant chromosome sizes.

In summary, although authors employed cutting edge technologies, the insufficiencies in experimental design and presentation preclude our understanding of the full scope of events.

Minor points:

The last sentence of the introduction, “Therefore, our results and methods -------” is by far beyond the scope of the study.

In Fig. 1, an arrow indicating a primer for the tail junction is unclear.

In the subtitle “2.4. Two ScURA3 insertion ------” “corruption” should be “degeneration.”

Author Response

Authors’ reply to Reviewer #1

Reviewer 1:

We are grateful for reviewer 1’s comments and responded accordingly. See below.

“The experimental design was somewhat insufficient to understand the full scope of the events. In particular, Nanopore results are not adequately presented. This reviewer would assume this is because of the lack of read depth of the Nanopore data. On average, 370Mb is read for each clone, which is approximately 26 fold coverage of the K. marxianus genome (14Mb, which was not mentioned in the manuscript). With the high error rate, 26x coverage is not sufficient to assemble the individual clones’ genome, and thus to reveal a full scope of insertion events.”

-the genome size of the K. marxianus (NRRL Y-6860) that we used was about 10.8 Mb (see Materials and Method section under 4.1 Yeast strains) Nanopore data is about 370Mb for each clone. It is about 34X coverage. If we were doing de novo sequencing of our K. marxianus genome, that coverage would be well below the recommended 45-50X normally used for Nanopore de novo genome sequencing. We already have a high quality PacBio sequence for our K. marxianus isolate ( GenBank assembly accession: GCA_002356615.1). One of our authors, Brad Abramson, who uses Nanopore sequencing for most of his work, assures us that our 34X coverage is sufficient to characterize the chromosome rearrangements we report in this paper.

which is above the standard 30X for Nanopore sequencing and in our opinion good for scope of the paper.

“Each clone is supposed to have a different number of integration events, which should be able to find out by estimating the copy numbers of Ura3 cassette. This can be done either by Southern blot or quantitative digital PCR. Then the authors could use Illumina short read paired-end sequencing for the entire genome (not for PCR products) and Nanopore sequencing for some of the clones that have high incidences of integration. In this way, authors could have had sufficient read depths for each sequencing platform and revealed a full scope of the events, which is necessary for publication.”

-Ura3 cassette copy number estimation with Southern blot or qPCR is definitely ideal. Yet our rough estimation (Figure S4) is adequate for the scope of the paper. We are not trying to get to know the 24 randomly-picked clones to the single base level, rather we are illustrating the gross genome rearrangement occurred in K. marxianus and the tools we used could help researchers understand such rearrangement better. Researchers have the discretion based on their means to generate more data/depth/coverage for their own studies in the field.

“In addition to the insufficient presentation of the Nanopore sequencing data, the CHEF data is confusing. Is this Southern blot with the ScUra3 as a probe, or a simple CHEF gel picture? If it is a simple CHEF gel picture, why some clones lack some of the chromosomes? Instead of proposing an interpretation, authors could blot the gel into the membrane and probe with ScUra3. In this way, the authors will know the aberrant chromosomes are indeed induced by the integration events. Additional whole genome sequencing and copy number analysis could tell whether there are deletions and amplifications that contribute to the aberrant chromosome sizes.”

-Figure 4 is a CHEF gel picture. The schematic on the left (Figure 4A) and text explains the details.

Minor points:

The last sentence of the introduction, “Therefore, our results and methods -------” is by far beyond the scope of the study.

We have softened the sentence some.

In Fig. 1, an arrow indicating a primer for the tail junction is unclear.

We do not see any problem with the Figure.

In the subtitle “2.4. Two ScURA3 insertion ------” “corruption” should be “degeneration.”

We changes this as suggested.

Reviewer 2 Report

The manuscript by Ding et al. “Gross Chromosomal Rearrangements in Kluyveromyces marxianus revealed by Illumina and Oxford Nanopore sequencing” explores the precise sequences at the junctions of Double Strand Breaks (DSB) generated by Non-Homologous End Joining (NHEJ) repair mechanism in Kluyveromyces marxianus. To obtain clones with new repair events, colonies of an uracil auxotrophic strain that had recovered the ability to synthesize uracil after inserting a Saccharomyces gen cassette were chosen. The use of NGS technologies allowed to determine the precise nucleotide sequence at the junctions of the insertion sites in the K. marxianus genome. The PCR tests revealed several insertions of tandem concatemers. Interestingly, six of the 24 selected clones showed gross chromosomal rearrangements (2 inversions, 1 transposition, and 3 translocations) requiring at least two DSBs per cell.

NHEJ is one of the DSBs repair mechanisms that acts to prevent structural damage to chromosomes. Therefore, it is important to know how it works to understand the processes in which it can fail, generating structural changes in the chromosomes. The knowledge of the origin of these changes in chromosomes is of great interest in various fields of research, from evolutionary genetics to clinical oncology

Comments to the authors:

I find the topic very interesting for researchers from different fields. In addition, the experiments carried out are solid, using an adequate and very powerful methodology. Overall, I consider the text to be correct and adequate, except for the discussion section which I think should be improved a bit.

  1. As stated above, my main concern is the discussion section, which I think that as it is now it looks more like a conclusions list that could be the end of the discussion. Some of the questions I raise bellow (2, 3, 7) could be explained in the discussion section. In addition, the last sentence of the discussion, “Further investigation may uncover DNA repair mechanisms that could be key to understanding cancer biology”, seems a bit forced to me. I am not a cancer biologist and, perhaps for that reason, I would have liked to find a little of discussion about the importance of knowledge of the DNA repair mechanisms for understanding cancer biology before ending with this statement. On the other hand, the DNA repair mechanisms and particularly the NHEJ of DSBs are of capital importance to understand the generation of chromosomal rearrangements like those observed in this paper. These structural changes in chromosomes can generate rapid evolutive changes that in some cases could even lead to speciation events. Thus, the results here shown are also of great interest to researchers of molecular evolution, not only to cancer researchers.

  1. As Table 1 shows, the insertion of the ScURA3 fragment in some of the clones falls within other genes. However, there are only comments on the effect of this in clone 1. What about the other insertions that also fall within genes? If no effects are observed, how could be it explained?

  1. I was surprised to find out that all the concatemers were joined in the same sense, head to tail. I wonder if there is any reason not to find any tail to tail, or head to head junctions.

  1. Table 1. Maybe I got lost, but I have two doubts: A) I do not understand why clone 1 has an asterisk if there is information for all columns. It is true that primers have not been described, but it seems that they have not been necessary since both the position and that it is a monomer were determined by NGS. B) How could you determine that clone 13 has a monomer insert? A set of primers to test insert size is not described and since the chromosome location of the insert was not found, I interpret that there is no Nanopore sequence available.

  1. Check carefully the sequences provided, specially the one in Figure S2. The part indicated as ScURA3 ORF is not an ORF. Its length is not a multiple of three, and it contains many STOP codons. I noticed these problems when I unsuccessfully searched for some of the primer sequences to better understand the experiments. In addition, in both line 222 and line 250 of the main text it says that Figure S2 shows 1121 bp while I only count 1115 bp.

  1. The results show the junction sequences between a particular ScURA3 gene and the K. marxianus genome. The experiments also uncovered some NHEJ events in which the ScURA3 gene seems not to be directly involved. These junctions could provide valuable information on the repair mechanisms without the intervention of external DNA fragments. I understand that in clone 15 it was not possible to locate the rejoin of chromosome 1 and 5 since the junction in chromosome 5 falls in a repetitive region. But what about clones 5 and 23 for translocations, clone 6 for the other end of inversion and clone 7 for the transposed fragment into chromosome 7?

  1. The CHEF gel experiments show that in clone 23 the band corresponding to chromosome 5 has disappeared. This is an unexpected result that the text describes as “interesting”. Nevertheless, no more mention about this is done. What could be the reason for this result?

  1. Please, verify the authorship because one of the authors, Joe Liang, does not appear in the author contribution section.

Minor comments and typos:

L21. The text says “We transformed a PCR product of the Saccharomyces cerevisiae URA3 gene (ScURA3) into an uracil auxotroph K. marxianus wildtype strain”. This could be a bit confusing as it is not a really wildtype strain (as it occurs in nature). The uracil auxotrophy of this strain is the result of an experimental modification (as explained in line 213). Authors may consider delete the “wildtype” word.

L 16 and L32 “Kluyveromyces marxianus (K. marxianus)” delete the repeated name in parentheses.

Line 32 Change “…is thermotolerant yeast and it the fastest…”to “…is a thermotolerant yeast and is the fastest…”

L 44. The meaning of DSB must be introduced the first time it is used.

L 50. Spell the last name of the cited author correctly. Change “Abdel-Banet” to “Abdel-Banat” as it appears in the cited article.

L 55 “This is unlikely to be achieved by Southern blot hybridization due to its low resolution, not to mention its laboriousness.” I would not say that “this is unlikely”, I would better say that “This is not possible”.

L57 “We generated our own Ura+ transformants…” please, indicate the species used. For example, “We generated our own Ura+ K. marxianus transformants…”

L 83 and L 95. Change “Table 1S” to “Table S1”.

L 117-118 Cross-check data in this phrase with that shown in Figure 2. “In clone 19, The ScURA3 tail had lost 25 terminal bases followed by 3 base substitutions (Figure 2).” In Figure 2, I count 24 lost bases plus 3 substitutions plus an initial G unclassified but that it looks like a substitution.

L 134. “there were no junction sequences in Table 1“. I think the text here should refer to Figure 2, not Table 1.

L 142 The meaning of CHEF must be introduced here.

Figure 4. What does “from a and B” mean? Maybe it could be deleted.

L153 Change “Figure 3a” to “Figure 3A”

L 179 Please, check the following sentence: “In addition, a short piece of chromosome 1 is released and integrates into chromosome 5 at approximately position 627037 (Figure 3B).” This fragment in Figure 3B, seems to belong to chromosome 4 (not 1), since the coordinates shown correspond exactly to the DSBs of chromosome 4.

L 210. Change “biologys to “biology”

L 215. “comprising the 10,837,618 genome” Indicate the units that correspond to that number (bp).

L 259. The primers “ura3+720c and ura3 R1-1-DN” do not appear in the Table S1.

Check the species names in the reference section. They should be in italics and also many of the genus names are written without capital letters (references 7, 8, 9, 10, 11 and 12).

Figures 1 and 3. The “H” and “T” labels are very difficult to see. The quality of the image should be improved or, better yet, the size of these labels should be increased

Figure S4 Use italics in the legend species name.

Table S1. Please, indicate why some letters are in lower case.

Author Response

Authors reply to Reviewer #2

We appreciate reviewer 2’s comments, suggestions and kind proofreading. We have responded accordingly. See below.

Comments and Suggestions for Authors

The manuscript by Ding et al. “Gross Chromosomal Rearrangements in Kluyveromyces marxianus revealed by Illumina and Oxford Nanopore sequencing” explores the precise sequences at the junctions of Double Strand Breaks (DSB) generated by Non-Homologous End Joining (NHEJ) repair mechanism in Kluyveromyces marxianus. To obtain clones with new repair events, colonies of an uracil auxotrophic strain that had recovered the ability to synthesize uracil after inserting a Saccharomyces gen cassette were chosen. The use of NGS technologies allowed to determine the precise nucleotide sequence at the junctions of the insertion sites in the K. marxianus genome. The PCR tests revealed several insertions of tandem concatemers. Interestingly, six of the 24 selected clones showed gross chromosomal rearrangements (2 inversions, 1 transposition, and 3 translocations) requiring at least two DSBs per cell.

NHEJ is one of the DSBs repair mechanisms that acts to prevent structural damage to chromosomes. Therefore, it is important to know how it works to understand the processes in which it can fail, generating structural changes in the chromosomes. The knowledge of the origin of these changes in chromosomes is of great interest in various fields of research, from evolutionary genetics to clinical oncology

Comments to the authors:

I find the topic very interesting for researchers from different fields. In addition, the experiments carried out are solid, using an adequate and very powerful methodology. Overall, I consider the text to be correct and adequate, except for the discussion section which I think should be improved a bit.

 “1. As stated above, my main concern is the discussion section, which I think that as it is now it looks more like a conclusions list that could be the end of the discussion. Some of the questions I raise bellow (2, 3, 7) could be explained in the discussion section. In addition, the last sentence of the discussion, “Further investigation may uncover DNA repair mechanisms that could be key to understanding cancer biology”, seems a bit forced to me. I am not a cancer biologist and, perhaps for that reason, I would have liked to find a little of discussion about the importance of knowledge of the DNA repair mechanisms for understanding cancer biology before ending with this statement. On the other hand, the DNA repair mechanisms and particularly the NHEJ of DSBs are of capital importance to understand the generation of chromosomal rearrangements like those observed in this paper. These structural changes in chromosomes can generate rapid evolutive changes that in some cases could even lead to speciation events. Thus, the results here shown are also of great interest to researchers of molecular evolution, not only to cancer researchers.”

-We tuned down the statement.

“2. As Table 1 shows, the insertion of the ScURA3 fragment in some of the clones falls within other genes. However, there are only comments on the effect of this in clone 1. What about the other insertions that also fall within genes? If no effects are observed, how could be it explained?”

-We were able to discover the insertion sites of 22 out of 24 clones. Clone 1 and 13 (* in table 1) were discussed in the text. Clone 13 was talked about in line 138-140.

“3. I was surprised to find out that all the concatemers were joined in the same sense, head to tail. I wonder if there is any reason not to find any tail to tail, or head to head junctions.”

-We randomly picked 24 clones for our study. And there are almost likely other interesting genome arrangement events that are yet to be discovered in this organism. In depth study of the pathway with loss-of-function and gain-of-function experiments will be extremely helpful understanding the mechanisms.

“4. Table 1. Maybe I got lost, but I have two doubts: A) I do not understand why clone 1 has an asterisk if there is information for all columns. It is true that primers have not been described, but it seems that they have not been necessary since both the position and that it is a monomer were determined by NGS. B) How could you determine that clone 13 has a monomer insert? A set of primers to test insert size is not described and since the chromosome location of the insert was not found, I interpret that there is no Nanopore sequence available.”

-Clone 1 (line 123-132) was asterisked since we could not find the source of the 65bp insertion. As for clone 13 the 20bp ScURA3 tail and 78bp head was the only remain we found. A monomer is an educated estimation. 

“5. Check carefully the sequences provided, specially the one in Figure S2. The part indicated as ScURA3 ORF is not an ORF. Its length is not a multiple of three, and it contains many STOP codons. I noticed these problems when I unsuccessfully searched for some of the primer sequences to better understand the experiments. In addition, in both line 222 and line 250 of the main text it says that Figure S2 shows 1121 bp while I only count 1115 bp.”

-Bold and un-italicized (from underscored orange ATG to TAA) is an ORF. The whole amplicon is 1123bp and we fixed it in the text and Fig S2.

“6. The results show the junction sequences between a particular ScURA3 gene and the K. marxianus genome. The experiments also uncovered some NHEJ events in which the ScURA3 gene seems not to be directly involved. These junctions could provide valuable information on the repair mechanisms without the intervention of external DNA fragments. I understand that in clone 15 it was not possible to locate the rejoin of chromosome 1 and 5 since the junction in chromosome 5 falls in a repetitive region. But what about clones 5 and 23 for translocations, clone 6 for the other end of inversion and clone 7 for the transposed fragment into chromosome 7?”

-We provided sequences in supplementary SupplementaryFigures.

“7. The CHEF gel experiments show that in clone 23 the band corresponding to chromosome 5 has disappeared. This is an unexpected result that the text describes as “interesting”. Nevertheless, no more mention about this is done. What could be the reason for this result?”

 -Based on the sequencing data, chromosome 5 should be at where it was indicated. Yet it is possible that the chromosome is experiencing some sort of rearrangement that renders it unable to enter the gel, therefore not able to show on the gel picture. A tweak of genome preparation for CHEF may resolve this. But the beauty of this paper is to show that the NGS sequencing (genome extraction) is filling the gap.

“8. Please, verify the authorship because one of the authors, Joe Liang, does not appear in the author contribution section.”

-We have corrected that. Thank you!

Minor comments and typos:

-We thank you sincerely for going into the details. We literally went through the bullet points and corrected the text and figures as you pointed out. You made it so easy for us to fix/polish the paper. Your efforts are truly precious in science and we really appreciate it.

-L21. The text says “We transformed a PCR product of the Saccharomyces cerevisiae URA3 gene (ScURA3) into an uracil auxotroph K. marxianus wildtype strain”. This could be a bit confusing as it is not a really wildtype strain (as it occurs in nature). The uracil auxotrophy of this strain is the result of an experimental modification (as explained in line 213). Authors may consider delete the “wildtype” word.

-L 16 and L32 “Kluyveromyces marxianus (K. marxianus)” delete the repeated name in parentheses.

-Line 32 Change “…is thermotolerant yeast and it the fastest…”to “…is a thermotolerant yeast and is the fastest…”

-L 44. The meaning of DSB must be introduced the first time it is used.

-L 50. Spell the last name of the cited author correctly. Change “Abdel-Banet” to “Abdel-Banat” as it appears in the cited article. 

-L 55 “This is unlikely to be achieved by Southern blot hybridization due to its low resolution, not to mention its laboriousness.” I would not say that “this is unlikely”, I would better say that “This is not possible”.

-L57 “We generated our own Ura+ transformants…” please, indicate the species used. For example, “We generated our own Ura+ K. marxianus transformants…”

-L 83 and L 95. Change “Table 1S” to “Table S1”.

-L 117-118 Cross-check data in this phrase with that shown in Figure 2. “In clone 19, The ScURA3 tail had lost 25 terminal bases followed by 3 base substitutions (Figure 2).” In Figure 2, I count 24 lost bases plus 3 substitutions plus an initial G unclassified but that it looks like a substitution.

-L 134. “there were no junction sequences in Table 1“. I think the text here should refer to Figure 2, not Table 1.

-L 142 The meaning of CHEF must be introduced here.

-Figure 4. What does “from a and B” mean? Maybe it could be deleted.

-L153 Change “Figure 3a” to “Figure 3A”

-L 179 Please, check the following sentence: “In addition, a short piece of chromosome 1 is released and integrates into chromosome 5 at approximately position 627037 (Figure 3B).” This fragment in Figure 3B, seems to belong to chromosome 4 (not 1), since the coordinates shown correspond exactly to the DSBs of chromosome 4.

-L 210. Change “biologys to “biology”

-L 215. “comprising the 10,837,618 genome” Indicate the units that correspond to that number (bp).

-L 259. The primers “ura3+720c and ura3 R1-1-DN” do not appear in the Table S1.

-Check the species names in the reference section. They should be in italics and also many of the genus names are written without capital letters (references 7, 8, 9, 10, 11 and 12).

-Figures 1 and 3. The “H” and “T” labels are very difficult to see. The quality of the image should be improved or, better yet, the size of these labels should be increased

-Figure S4 Use italics in the legend species name.

-Table S1. Please, indicate why some letters are in lower case.

Reviewer 3 Report

I read the manuscript of Ding et al., entitled “Gross Chromosomal Rearrangements in Kluyveromyces marxianus revealed by Illumina and Oxford Nanopore sequencing” with great interest. The authors explore the effect of cellular DNA repair mechanisms (e.g. NHEJ) on provoking gross chromosomal rearrangements. The authors used the yeast K. marxianus to artificially insert a PCR product of the URA3 gene, and then they sequence the genome of 24 transformed clones, to detect the effect of the URA3 insertions. The manuscript is well written and a pleasure to read, the methodology and the results are convincing, and the discussion is short but accurate. In this context, I recommend it for publication in IJMS.

Author Response

We thank reviewer 3 for the kind words.

Round 2

Reviewer 1 Report

The authors stated in the abstract that "However, the precise DNA sequence at the insertion sites were not fully explored." I agree with the author that de novo assembly would be an excellent way to fully determine the events in each clone. Organisms with the small genome, such as K. marxianus would be feasible. Knowing exactly how many copies are integrated into the genome in each clone would help.